# Effects of Contact Conditions between Particles and Volatiles during Co-Pyrolysis of Brown Coal and Wheat Straw in a Thermogravimetric Analyzer and Fixed-Bed Reactor

**Lingmei Zhou [1], Guanjun Zhang [2],\*, Ligang Zhang [3], Denise Klinger [4] and Bernd Meyer [4]**

[1] School of Chemical & Environmental Engineering, China University of Mining and Technology (Beijing), D11, Xueyuan Road, Haidian District, Beijing 100083, China; lingmeizhou@hotmail.com

[2] China Datang Corporation Science and Technology Research Institute, Beinong Road 2, Changping District, Beijing 100083, China

[3] School of Materials Science and Engineering, Central South University, Changsha, Hunan 410083, China; ligangzhang@csu.edu.cn

[4] TU Bergakademie Freiberg, Institute of Energy Process Engineering and Chemical Engineering (IEC) and German Center for Energy Resources (DER), Fuchsmühlenweg 9, 09599 Freiberg, Germany; denise.klinger@iec.tu-freiberg.de (D.K.); bernd.meyer@iec.tu-freiberg.de (B.M.)

\* Correspondence: guanjun2014@hotmail.com; Tel.: +86-185-1857-7101

**Abstract:** Biomass is a clean and renewable energy source. In order to partially replace fossil fuels and break the limitations of the usage of biomass alone, co-pyrolysis of coal and biomass has been increasingly focused on by researchers, but few articles have investigated the effects of contact conditions between volatiles produced by thermal decomposition and particles from each other during co-pyrolysis. In the present work, co-pyrolysis behavior of wheat straw (WS) and brown coal (HKN) was investigated in a thermogravimetric analyzer (TGA) up to 1100 °C and a fixed bed reactor up to 800 °C, with different contact conditions of particles (biomass was placed above, below and well-mixed with coal). The results showed that the most obvious interactions occurred for mixed sample when 10 wt.% biomass was placed below the coal and mixed sample when 50 wt.% biomass was placed above the coal, both in TGA and fixed bed reactor, with different mechanisms. The synergy effect related to interactions that occurred during co-pyrolysis lead to different behaviors compared to simply addition of coal and biomass: In TGA this was caused by longer reaction time between particles and volatile products produced in primary pyrolysis process. However, for the fixed bed reactor, much more volatiles and catalytic compounds were produced to promote the particles decomposition of WS and HKN. Therefore, opposite to TGA, obvious synergy effects occurred for the blend with less contact time and were caused by volatiles containing more $H_2$ and catalytic materials, which reacted with particles of the other fuel species along the gas flow direction. The kinetic parameters obtained by the Coats–Redfern method agreed with experimental behaviors and synergy effects.

**Keywords:** co-pyrolysis; contact conditions; wheat straw; brown coal

## 1. Introduction

Rapid economic growth requires a lot of energy, most of which currently still comes from fossil energy. However, the use of fossil energy brings a series of problems, such as environmental pollution, non-renewable characteristic and so on. Nowadays, many countries focus on green renewable energy sources such as wind, solar [1] and biomass energy. The utilization of "green energy" biomass has

received worldwide attention. Biomass is a clean and renewable energy source and it is generally accepted as one which would not exacerbate the greenhouse effect in the atmosphere. It can absorb $CO_2$ and transform it into organic matter and oxygen. When using biomass for thermal processing, such as combustion or pyrolysis, it generates as much $CO_2$ as it absorbs, so the whole process does not increase the amount of $CO_2$ in the atmosphere [2–6]. In addition, compared to coal biomass it has a high content of volatile matter, hydrogen and catalytic minerals, leading to a higher thermal reactivity. Therefore, biomass is more suitable for producing liquid fuels and other chemical products. However, biomass utilization has some problems: The supply is unsteady (seasonal) and the energy density is low (low bulk density, low heating value, high water content), which results in difficulties with storage due to high space requirements and biological degradation. These facts limit the feasibility of biomass as a single raw material. In order to partially replace fossil fuels and break the limitations of the usage of biomass alone, many researchers deal with thermal co-utilization of coal and biomass [7–11]. Among these co-utilizations, co-pyrolysis of coal and biomass was paid wide attention, since pyrolysis is the initial stage for combustion and gasification.

Co-pyrolysis of coal and biomass has been investigated by many studies [12–19]. However, the findings were always conflicting, indicating some underlying processes occurring during the co-pyrolysis process. Some researchers confirmed synergy effects during co-pyrolysis process regarding product yields, gas components or decomposition rates [16–19]. These synergy effects were mainly attributed to higher content of catalytic-active mineral matter in biomass, especially alkali and alkaline earth metal species. In addition, it can be found from literature that synergy effects were more likely to occur in large reactors (i.e., fixed bed reactor) compared to relatively small ones (i.e., traditional thermogravimetric analyzer—TGA) [14,15]. Sonobe et al. [14] investigated the co-pyrolysis behavior of Thai lignite with corncob and found no obvious interactions using TGA but significant interactions in a fixed bed reactor. Dong et al. [15] studied co-pyrolysis of sawdust and coal in a TGA and fixed bed reactor and reported that interactions mainly occurred from 400 to 700 °C in TGA compared to 500 to 700 °C in a fixed bed reactor. The different behaviors might be caused by different residence time in large and small reactors. Zhu et al. [20] observed an increased gas product yield from 21.3 to 37 wt.% and a corresponding decreasing char yield (from 17.4 to 8.03 wt.%) when residence time increased from 8.5 to 34 s in co-pyrolysis. On the contrary, additive behavior, which means no interactions occurred during co-pyrolysis and the experimental behavior can be obtained by simple addition between coal and biomass, occurred during co-pyrolysis of coal and biomass [11,12]. It was mainly explained by the fact that the volatiles were quickly swept away by a high flow of the carrier gas, thus volatile–char interactions were hindered.

As a conclusion of the previous literature findings, it can be said that the occurrence of synergy effects is related to secondary volatile–char or volatile–volatile interactions. Therefore, it is assumed that the contact conditions between volatiles and particles should be significant to the pyrolysis process, besides the temperature as a main influencing factor in secondary reactions. Only a few articles have studied the effects of contact conditions during co-pyrolysis of coal and biomass. Zhang et al. [21] studied co-gasification of coal and biomass in a fixed bed reactor with separate and mixed bed configurations and observed that the pre-mixed bed configuration produced a well-dispersed bio-ash among coal char grains, which indicated stronger synergy than the separate ones. The authors investigated the contact conditions of coal and biomass particles, but only focused on the bio-ash distributions, without paying attention to the volatiles' diffusion path in the blend samples. This paper focused on this missing information in the literature.

In the present work, co-pyrolysis behavior of wheat straw (WS) and brown coal (HKN) was investigated in a thermogravimetric analyzer (TGA) and a fixed bed reactor (LPA: laboratory pyrolysis apparatus), with different contact conditions of the two materials applied. The aim was to find out how the contact conditions between particles and volatiles influence synergy effects and, therefore, the pyrolysis process. Three different contact conditions of particles were considered: (a) Well-mixed, (b) biomass placed above coal and (c) biomass placed below coal. The experiments were carried out at

a constant heating rate of 10 K/min at atmospheric pressure, while the temperature varied between the fixed bed reactor LPA (25–800 °C) and TGA (25–1100 °C). Based on the experimental data from TGA, the kinetic parameters (activation energy and the frequency factor) for the total volatile released in case of single and blend fuels were determined using the Coats–Redfern method [22,23].

## 2. Materials and Methods

### 2.1. Materials

Three kinds of samples were used in this study: Wheat straw (WS), a brown coal from open cast-mine Hambach in the Rhenish lignite mining region (HKN), and their blends in different ratios (10 wt.% and 50 wt.% WS addition based on dry samples). The raw materials were pre-dried and milled to get the particle size to less than 2 mm. All samples were oven-dried at 105 °C and maintained at this temperature for 24 h to get rid of moisture. The characteristics of raw wheat straw and brown coal were performed according to the related DIN standards (Germany). The ultimate, proximate and ash analysis of each raw sample are shown in Table 1. The composition of blend samples can be calculated by the rule of mixing.

**Table 1.** Ultimate, proximate and ash analysis of raw brown coal (HKN) and wheat straw (WS).

| Sample | Wheat Straw (WS) | Rhenish Brown Coal (HKN) |
|---|---|---|
| **Ultimate Analysis (wt.%), d** | | |
| Carbon, C | 49.26 | 69.04 |
| Hydrogen, H | 5.98 | 5.01 |
| Nitrogen, N | 0.67 | 0.79 |
| Sulfur, S | 0.30 | 0.64 |
| Oxygen, O (diff [a]) | 43.79 | 24.52 |
| **Proximate analysis (wt.%), d** | | |
| Moisture [b] | 10.93 | 51.12 |
| Ash | 6.88 | 5.47 |
| Volatile matter | 75.85 | 50.70 |
| Fixed Carbon | 17.27 | 43.83 |
| **Ash analysis [c] (wt.%), d** | | |
| $Na_2O$ | 0.42 | 6.34 |
| $MgO$ | 2.24 | 16.21 |
| $K2O$ | 15.82 | 0.91 |
| $CaO$ | 6.71 | 37.51 |
| $Fe_2O_3$ | 0.40 | 10.13 |
| $Al_2O_3$ | 0.81 | 3.90 |
| $SiO_2$ | 62.79 | 9.39 |
| Index of basicity [e] | 0.03 | 0.29 |

[a] Calculated by difference; [b] based on received sample; [c] only oxides in index of basicity are shown here; [d] dry sample; [e] index of basicity $= w(A) * \frac{Fe_2O_3 + CaO + MgO + Na_2O + K_2O}{SiO_2 + Al_2O_3}$.

### 2.2. Methods

#### 2.2.1. Pyrolysis Experiments

(1) Thermogravimetric analyzer (TGA)

The thermogravimetric analyzer (TGA) used in this study was manufactured by Deutsche Montan Technologie (DMT) Co. in Freiberg, Germany. As shown in Figure 1, the system mainly consists of a micro-balance connected with the sample basket by a removable metal chain, the reactor vessel, an electrically heated furnace, the gas supply system and the data collection system. For a pyrolysis run, the sample was firstly put into a specially designed sample holder. This sample holder/basket

consists of a wire mesh (height of the mesh: 2 mm) and two metal caps (diameter: 1 cm) at the bottom and top of it. This sample basket was connected with the micro-balance via a metal chain on the top. It is assumed that the produced volatiles can be released from the sample via the wire mesh. For the non-isothermal experiments in this study, the basket (including the sample) was firstly dropped down into the reactor by a cable winch. A constant $N_2$ stream of 1 L/min (STP) was used throughout the process to achieve an inert atmosphere and to remove the released volatiles from the hot reaction zone. Non-isothermal pyrolysis experiments were carried out at a heating rate of 10 K/min from room temperature to 1100 °C, at atmospheric pressure and with wheat straw (WS), brown coal (HKN) and their blends (10, 50 wt.% of WS) in the three mentioned contact conditions. The sample weights were in the range of 0.26–0.61 g, while the sensitivity of the micro-balance is 1 µg. The repeatability and reproducibility were found to be good for samples, at a 99% confidence level.

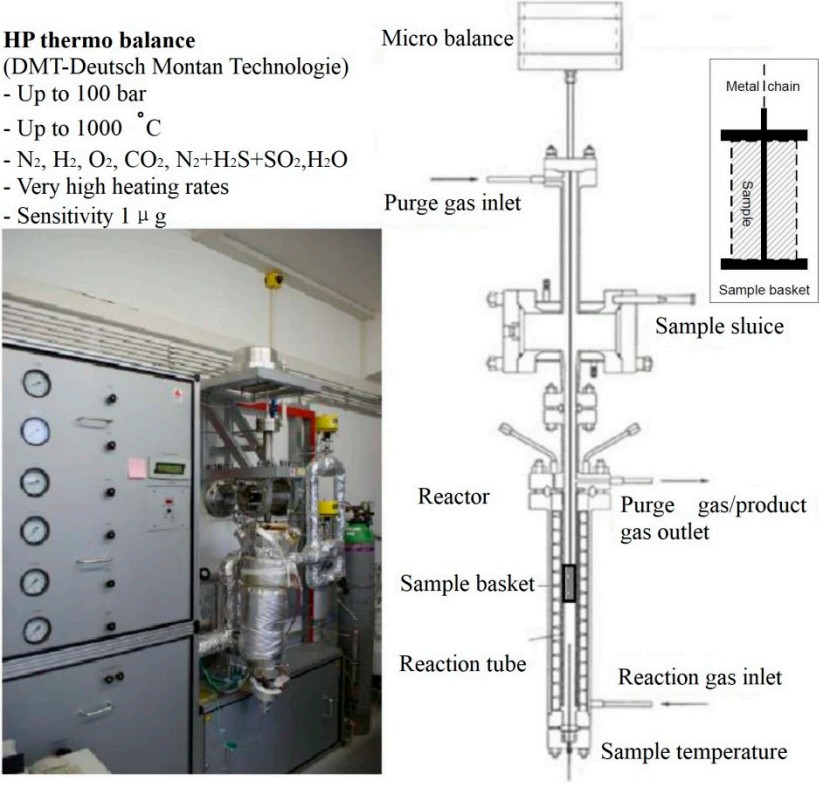

**Figure 1.** Schematic of thermogravimetric analyzer (TGA).

(2) Fixed bed reactor (LPA)

A fixed bed reactor named LPA (laboratory pyrolysis apparatus) was used to obtain the product yields of char, liquids and gas as well as the product composition from the pyrolysis of wheat straw, brown coal and their blend samples. A scheme of the experimental setup is shown in Figure 2. The LPA reactor was heated by a vertically movable tube furnace, which was electrically heated. The sample temperature was measured by a thermocouple placed inside the reactor in the bulk sample. The sample was put inside the reactor and some glass wood was placed at the bottom and top of the reactor in order to avoid blocking of the gas tube. A constant stream of argon was used throughout the whole process to achieve an inert atmosphere and also to assist removal of the pyrolysis products. The total liquid product was trapped in two condensers/cold traps and all the non-condensable volatiles (gases) were firstly collected in six glass bottles (1 L), which were full of NaCl-solution and connected by rubber hose. After the experiment, the gases were collected in gas bags by lifting the tanks with NaCl-solution connected with every gas bottle. Then, all the gas bags were analyzed by a micro gas chromatograph (micro-GC) after experiment. The gas can be collected at a certain time or temperature interval by

controlling the communication switch between the glass bottles. The experiments were carried out using WS, HKN and their blends (10, 50 wt.% of WS), with an amount of 20 g. The samples were heated up from 25 to 800 °C with a heating rate of 10 K/min under argon atmosphere (50 mL/min STP).

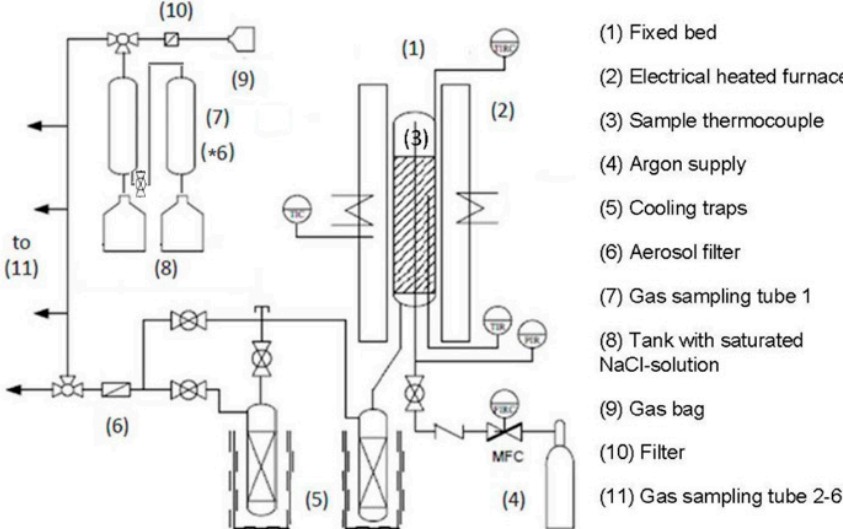

**Figure 2.** Schematic of fixed bed reactor (LPA).

2.2.2. Evaluation of Synergy Effects during Co-Pyrolysis

To investigate the synergy effect during co-pyrolysis of coal and biomass, the experimental derived parameters should be compared to the calculated ones. The latter are obtained by applying the rule of mixing. If the experimental behavior can be predicted well by the calculation according to the rule of mixing, the behavior during co-pyrolysis follows the so-called additive model:

$$(Y)_{blend} = x_1(Y)_{coal} + x_2(Y)_{biomass} \tag{1}$$

where Y stands for a given parameter related to pyrolysis, for example, the sample weight loss, the reaction rate, or the kinetic parameters, and $X_1$ or $X_2$ represents the coal and biomass ratio.

To investigate the degree of synergy effects during co-pyrolysis process, $\Delta Y$ is introduced as Equation (2):

$$\triangle Y = Y_{experimental} - Y_{calculated} \tag{2}$$

where $\triangle Y$ is the difference between the calculated and the experimental values, which can be assumed as an indicator of interaction.

2.2.3. Kinetics of Pyrolysis

The Coats–Redfern (CR) method [22,23] was applied to the TGA in this study to obtain the total volatile release kinetics for pyrolysis of wheat straw, brown coal and their blends. Under a constant heating rate β, the fundamental Arrhenius rate expression can be rearranged to

$$\frac{d\alpha}{dT} = \frac{A}{\beta} \cdot \exp\left(-\frac{E_a}{RT}\right) \cdot f(\alpha) \tag{3}$$

where $\alpha$ is the conversion ranging from 0 to 1, T is the temperature, R is the universal gas constant, A is the frequency factor and $E_a$ is the activation energy of the reaction. The model $f(\alpha)$ used here is expressed as

$$f(\alpha) = (1 - \alpha)^n \tag{4}$$

where n is the order of reaction. In many applications, the pyrolysis of fuels was assumed to be a first-order reaction (n = 1), which is related only to the decomposition reactions. Assuming n = 1, Equation (4) can be rearranged under a constant heating rate β to

$$\frac{d\alpha}{dT} = \frac{A}{\beta} \cdot \exp\left(-\frac{E_a}{RT}\right) \cdot (1 - \alpha) \tag{5}$$

The Coats–Redfern integral method in Equation (5) uses a Taylor series expansion to yield the following expression:

$$\ln\left(\frac{-\ln(1 - \alpha)}{T^2}\right) = \ln\left[\frac{AR}{\beta E_a}\left(1 - \frac{2RT}{E_a}\right)\right] - \frac{E_a}{RT} \tag{6}$$

Equation (6) can be simplified by recognizing that for customary values of $E_a$, the term $2RT/E_a \ll 1$, a straight line can be obtained from single heating rate data by plotting $\ln[g(\alpha)/T^2]$ versus $-1/T$. The activation energy $E_a$ can be derived from the slope of the line $E_a/R$ and the frequency factor A can be obtained from its intercept $\ln(AR/\beta E_a)$.

## 3. Results and Discussion

### 3.1. Characteristics of Raw Samples

#### 3.1.1. Sample Properties of Coal and Biomass

The characteristics of the used brown coal HKN and wheat straw WS samples (Table 1) showed the well-known differences like higher volatile matter and corresponding low fixed carbon content, as well as low carbon and high oxygen content for wheat straw. The ash content was in a similar range for both fuels, but ash composition differed. Inherent mineral matter, originally present in the carbonaceous matrix, is supposed to act as a catalyst during pyrolysis reactions. This is mainly attributed to the alkali and alkaline earth metals [24,25] such as potassium, sodium, calcium [26,27] and iron [28]. As shown in Table 1, HKN contained a significantly higher calcium (37.51 wt.%) and iron (10.13 wt.%) content, while the predominantly catalytic matter in biomass was potassium (15.82 wt.%), which was regarded as primary explanation for synergy effects during the co-pyrolysis process in the literature.

#### 3.1.2. Pyrolysis Behaviors of Single Fuels in TGA

Pyrolysis behaviors of single fuels at 10 K/min from 200 to 1100 °C and atmospheric pressure are given in Figure 3, by utilizing mass loss (TG) and reaction rate (DTG) plots.

According to the TG curves in Figure 3a, the thermal decomposition of wheat straw and Rhenish brown coal differed from each other. It can be seen clearly from the TG curves that WS produced much more volatiles compared to HKN, which was attributed to the higher volatile matter content of WS shown in Table 1. Reaction rate DTG plots of wheat straw and brown coal are shown in Figure 3b. It can be found that (1) the maximum reaction rate at the peak of wheat straw was much higher, almost seven times that of brown coal; (2) wheat straw reached its maximum peak at 345 °C, much earlier than the 443 °C for brown coal sample; and (3) there was a small peak for brown coal in 890 °C, which might be caused by $CO_2$ release from carbonate (HKN contained much calcium). The large difference of pyrolysis behavior between wheat straw and brown coal might cause a synergy effect in co-pyrolysis process.

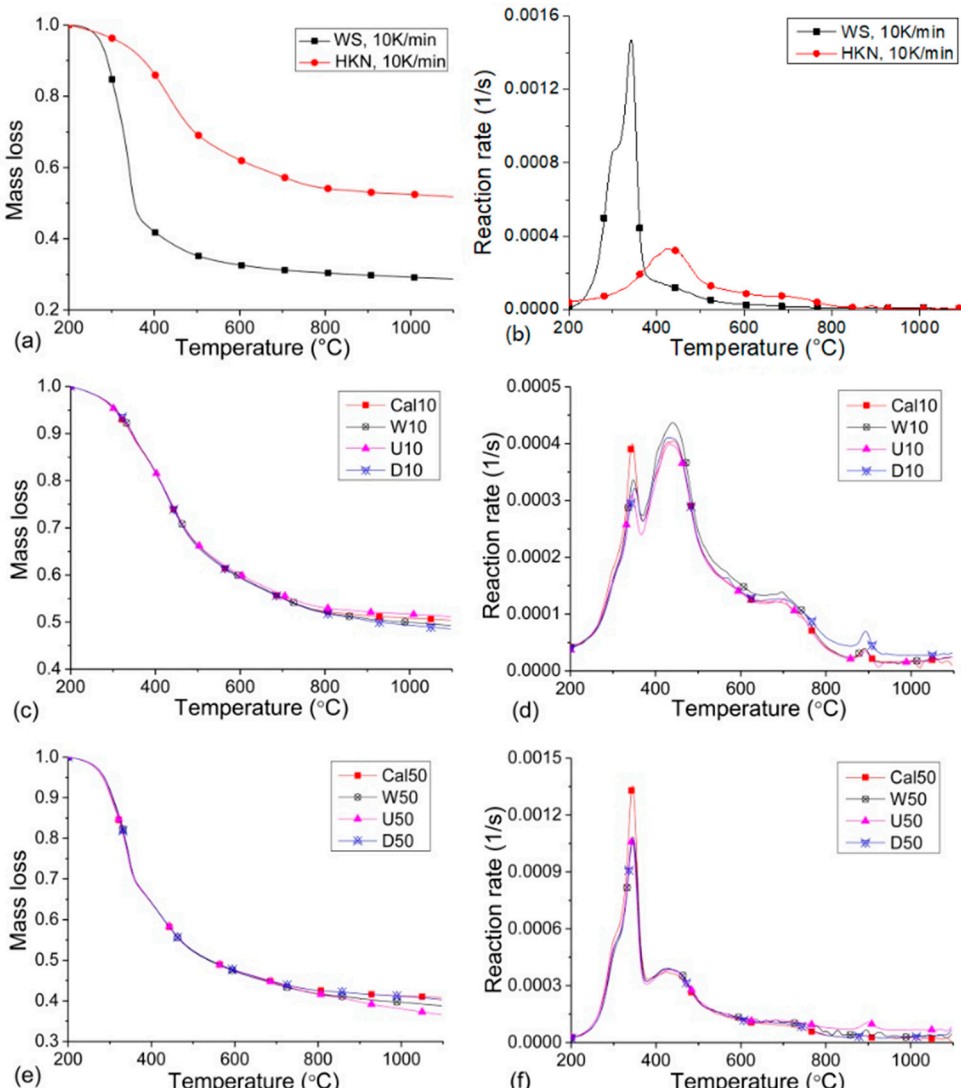

**Figure 3.** Pyrolysis behavior of single and blended samples: (**a**) mass loss (TG), single fuels, (**b**) reaction rate (DTG), single fuels, (**c**) TG, 10 wt.% samples, (**d**) DTG, 10 wt.% samples, (**e**) TG, 50 wt.% samples, (**f**) DTG, 50 wt.% samples.

### 3.2. Co-Pyrolysis of Wheat Straw and Brown Coal in TGA

As discussed in the literature, gas residence time (related to particle contact conditions in the present study) had strong effects on pyrolysis process, especially on the secondary pyrolysis reactions. Therefore, co-pyrolysis experiments were carried out in TGA with different contact conditions:

1) Biomass WS was placed above the HKN (up position), U10, U50;
2) Biomass WS was placed below the HKN (down position), D10, D50;
3) Well mixed samples were filled into the sample basket, W10, W50.

Sample symbols in the brackets can be explained as follows: U represents WS is in the up position; D represents WS in the down position; W shows WS and HKN are well distributed; numbers 10 and 50 correspond to 10 and 50 wt.% of wheat straw addition. The behaviors of blended samples were compared to calculated values (Cal10 and Cal50) from simple of coal and biomass. Cal10 represents the mixed sample which contained 10 wt.% wheat straw, and Cal50 represents the mixed sample which contained 50 wt.% wheat straw.

3.2.1. Mass Loss (TG) for Blend Samples

The mass loss curves (TG) for 10 wt.% and 50 wt.% of WS addition with different contact conditions are shown in Figure 3c,e and are compared to calculated values (in red). It can be seen that (1) in general, the differences in the mass loss curves compared to the calculated values occurred mainly above 600 °C; (2) when the temperature was above 600 °C, U10 released less volatile while W10 and D10 led to a higher volatile release; and (3) the mass loss of D50 was almost the same as expected, whilst the mass loss of W50 and U50 was higher than expected, and the volatile released by U50 was even higher than that of W50. It can be deduced that contact conditions had different effects according to the WS addition amount. The strongest synergy effects (highest difference between experimental and calculated values) for the volatile release occurred for the samples D10 and U50 (Δ yield for D10: 1.7 wt.%, U50: 4.2 wt.%), and in most cases the mass loss was higher than the calculated one.

3.2.2. Reaction Rate for Blend Samples

The reaction rate curves for wheat straw addition of 10 wt.% and 50 wt.% at different contact conditions are shown in Figure 3d,f. The profiles for the blended samples show differences in behavior depending on the WS addition ratio.

In case of 10 wt.% of WS addition, the curve displays two distinct peaks and a broader one: (1) The first peak showed around 340 °C, which was the position of the maximum decomposition peak for wheat straw. The reaction rate R1 for the first peak was obviously lower than calculated value. (2) The second peak showed around 440 °C, corresponding to the maximum peak for brown coal decomposition. The reaction rate of W10 and D10 was higher than the calculated value, while the reaction rate of U10 was slightly less than the calculated value. (3) For temperatures higher than 600 °C, the most obvious synergy effect occurred for D10 with an obvious higher decomposition rate, while U10 and W10 had almost the same reaction rate as calculated value.

When 50 wt.% WS was added to the brown coal HKN, the temperatures of the peaks and reaction rates changed significantly. There are two distinct peaks and a peak shoulder existent: (1) A shoulder around 300 °C related to the decomposition of hemicellulose in WS. The reaction rate for the shoulder was lower than calculated value. (2) A distinct peak at around 340 °C corresponding to the decomposition of cellulose in WS was much lower than expectation, showing remarkable synergy effect. (3) A peak around 440 °C assigned for primary pyrolysis stage for HKN, which was slightly higher than the calculated value. (4) The most obvious synergy effects occurred for U50 when the temperature was higher than 600 °C with a high decomposition rate.

As already observed for the mass loss curves, the interactions between brown coal and biomass varied with the contact conditions as well as with the temperature range. The most intensive interactions in the reaction rate took place between 200 and 375 °C for all samples with an obviously lower reaction rate than calculated, and at high temperatures (>600 °C) for D10 and U50 with obvious higher reaction rates. The latter effect was consistent with the synergy effects observed for the mass loss curves.

3.2.3. Explanations of Synergy Effects for Co-Pyrolysis in TGA

To interpret the experimental behavior, the hypothetical particle distribution (black: Rhenish brown coal, textured: Wheat straw) for blends, gas flow directions of $N_2$ and product volatiles are given in Figure 4. The experimental behaviors can be explained as follows:

1) Synergy effects at low temperature (<400 °C)

The lower reaction rate observed for the first peak showed at around 325 to 340 °C and was mainly related to heat and mass transfer limitations during the co-pyrolysis process. This temperature range is related to the cellulose decomposition from WS, which leads to a high amount of condensable volatile. When biomass was combined with coal particles, these peaks might be shifted to adjacent temperature intervals because of mass and heat transfer limitations. Haykiri et al. [29] investigated

the deviation of the devolatilization yields from theoretical values and found negative values around 230 °C. It can be also caused by the partially pyrolyzed brown coal particles, containing Ca, Fe and Na. Yang et al. [30] found the addition of minerals containing K, Na, Ca, Fe and Al, respectively, led to a decrease in hemicellulose and cellulose decomposition. With the comparison of kinetic parameters in Section 3.4, the mentioned situation can be explained more clearly.

2) Synergy effects at medium temperature (420 to 440 °C)

The higher reaction rates occurred between 420 and 440 °C and corresponded to the maximum decomposition rate for brown coal HKN, which was mainly caused by the catalytic effect brought by catalytic materials in wheat straw. Since the temperature was not high enough to release the catalytic materials in WS, the catalytic effect would occur only when the particles of WS and HKN contacted each other. Therefore, the most obvious synergy effect was for W10, in which WS particles distributed uniformly.

3) Synergy effects at high temperature (>600 °C)

As described above, the most significant interactions occurred for D10 (10 wt.% WS at the bottom) and U50 (50 wt.% WS at the top) when temperature was above 600 °C. However, the contact conditions of D10 and U50 differed as shown in Figure 4a,d. In TGA, the interactions for D10 were mainly caused by secondary pyrolysis among WS volatiles and catalytic active and partially decomposed HKN particles ($V_{WS} + P_{HKN}$). Similarly, for U50, the interactions should be mainly due to the secondary pyrolysis among HKN volatiles and WS particles ($V_{HKN} + P_{WS}$). The mechanisms for synergy effects of D10 and U50 seem to be totally different. As shown in Figure 4, the volatile releasing path in TGA was down–up, and the volatile residence time should depend on the volume of the particles at the upper position. The particle volume at the upper position for D10 (or U50) was much more than that for U10 (or D50), leading to longer contact time between volatiles and particles. As illustrated in literature, the volatile residence time had a significant influence on secondary pyrolysis. The most significant synergy effects which occurred for D10 and U50 samples might have been caused by longer contact time between particles and volatiles at their positions.

Therefore, based on the above explanation, it can be deduced that for blended samples W10 and W50, the higher mass loss was mainly caused by higher decomposition rate at a temperature range between 420 and 440 °C, which was caused by catalytic materials from wheat straw (WS) to promote the decomposition of brown coal (HKN).

For blended sample U10 and D50, the reaction rates at high and medium temperature were almost the same as calculated values, and the lower mass loss was mainly caused by lower decomposition rate of wheat straw in temperature range lower than 400 °C.

For blended sample D10 and U50, the higher mass loss was mainly caused by interactions between volatiles and particle of WS and HKN, which promoted decomposition at temperature higher than 600 °C.

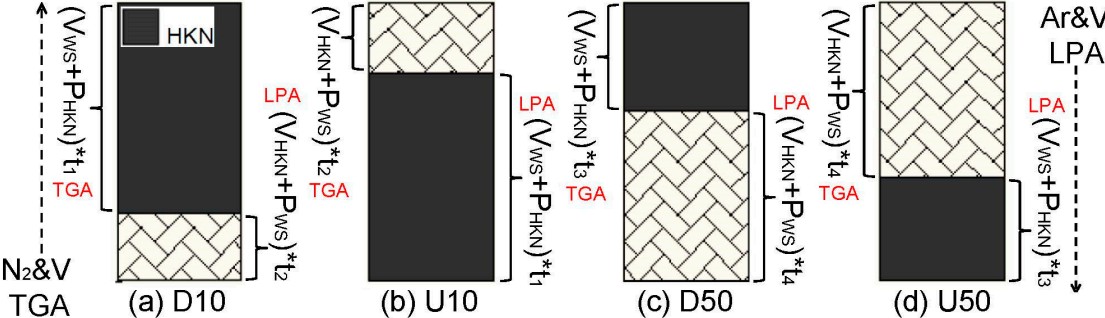

**Figure 4.** Mechanism for synergy effect occurred in TGA and LPA reactor under different contact conditions, (**a**) D10, (**b**) U10, (**c**) D50, (**d**) U50.

## 3.3. Co-Pyrolysis of Wheat Straw and Brown Coal in LPA

In this section, pyrolysis behavior of WS, HKN and their blends in the LPA reactor up to 800 °C are described, by regarding the occurrence of synergy effects on products yield and properties. Samples from WS and HKN were placed into the LPA reactor in three positions, as in TGA. However, it should be noted that the volatiles motion path in LPA was the opposite to that in the TGA. Argon was introduced from the top of the LPA reactor by metal tube and left with volatiles at the bottom of reactor. Furthermore, volatile residence times in the LPA reactor might differ from the TGA experiments because of different reactor dimensions, purge gas flow and sample height. Therefore, the synergy effects presented during co-pyrolysis process in the LPA reactor might be different compared to those from the TGA.

### 3.3.1. Char Characteristics in Fixed Bed Reactor (LPA)

1) Char yields of blend samples

Table 2 illustrates the experimental char yields compared to calculated values at different positions. The biggest difference occurred for D10 (3.30 wt.%) and U50 (2.00 wt.%), which showed lower char yields than the calculated values.

2) Characteristics of blend chars

The characteristics of chars pyrolyzed in LPA reactor including proximate analysis, ultimate analysis and ash composition are shown in Table 2, compared with the calculated values by the rule of mixing. For ultimate analysis, the carbon contents were approximately and slightly higher; the hydrogen and nitrogen contents were almost the same as the calculated ones; and the oxygen contents were much higher than expected, which indicated the reduction of oxygen-contained volatiles during the co-pyrolysis process. For ash composition analyzed by XRF, the content of potassium and calcium of all chars were lower than calculated values, which could cause a lower reactivity of the char samples. Besides, it can also indicate that some catalytic compounds (mainly K) were released into volatiles, which could promote the decomposition reaction during co-pyrolysis process.

**Table 2.** Char yields and characteristics for co-pyrolysis of WS and HKN in the LPA reactor compared to calculated values, at 800 °C.

| Sample | Cal.10% | W1 | U1 | D1 | Cal.50% | W5 | U5 | D5 |
|---|---|---|---|---|---|---|---|---|
| Char yield (wt.%) | 50.40 | 50.50 | 48.20 | 46.70 | 39.50 | 38.80 | 37.50 | 38.10 |
| Proximate and ultimate analysis (d, wt.%) | | | | | | | | |
| Volatile matter | 6.42 | 7.96 | 6.42 | 6.24 | 6.61 | 5.94 | 5.79 | 6.00 |
| Ash | 9.93 | 10.24 | 8.48 | 8.38 | 14.36 | 13.67 | 13.38 | 13.01 |
| C | 86.26 | 87.15 | 87.49 | 87.74 | 80.49 | 82.25 | 82.39 | 82.93 |
| H | 0.93 | 0.86 | 0.92 | 0.83 | 0.81 | 0.78 | 0.82 | 0.75 |
| N | 0.89 | 0.83 | 0.87 | 0.81 | 0.89 | 0.82 | 0.89 | 0.79 |
| S | 0.67 | 0.52 | 0.47 | 0.42 | 0.56 | 0.43 | 0.41 | 0.40 |
| O | 0.75 | 1.43 | 1.76 | 1.82 | 1.06 | 2.05 | 2.11 | 2.12 |
| Ash composition (d, wt.%) by XRF | | | | | | | | |
| $Na_2O$ | 0.71 | 0.70 | 0.70 | 0.73 | 0.46 | 0.57 | 0.50 | 0.52 |
| MgO | 1.03 | 1.06 | 1.05 | 1.07 | 0.80 | 0.87 | 0.87 | 0.89 |
| $Al_2O_3$ | 0.25 | 0.30 | 0.26 | 0.26 | 0.21 | 0.23 | 0.21 | 0.22 |
| $SiO_2$ | 2.05 | 2.09 | 1.54 | 1.48 | 8.18 | 6.61 | 6.42 | 6.01 |
| $P_2O_5$ | 0.06 | 0.03 | 0.02 | 0.02 | 0.28 | 0.19 | 0.19 | 0.19 |
| $SO_3$ | 1.68 | 1.30 | 1.18 | 1.05 | 1.39 | 1.08 | 1.03 | 1.00 |
| Cl | 0.05 | 0.07 | 0.08 | 0.05 | 0.08 | 0.08 | 0.08 | 0.06 |
| $K_2O$ | 0.39 | 0.27 | 0.28 | 0.21 | 1.71 | 1.23 | 1.31 | 1.26 |
| CaO | 3.98 | 3.00 | 3.00 | 3.02 | 3.01 | 2.59 | 2.55 | 2.60 |
| $TiO_2$ | 0.02 | 0.02 | 0.02 | 0.02 | 0.02 | 0.02 | 0.02 | 0.02 |
| $Fe_2O_3$ | 0.88 | 0.83 | 0.80 | 0.84 | 0.55 | 0.57 | 0.56 | 0.58 |
| BaO | 0.03 | 0.03 | 0.02 | 0.03 | 0.02 | 0.02 | 0.02 | 0.02 |

3.3.2. Gas Composition in Fixed Bed Reactor (LPA)

1) Gas composition from single pyrolysis of HKN and WS

The system for gas collection consists of six glass bottles with scale and the volume of each bottle was 1000 mL. The volume of gas products in each gas bag can be read by the scale on the bottle. By analysis of gas composition in each gas bag via a micro gas chromatograph, the total volume of certain gases (such as $H_2$, $CO_2$, CO) collected during the whole pyrolysis process can be calculated. Finally, the volume percent of certain gas can be obtained, as shown in Figure 5. The comparison of product gas composition (vol.%) from wheat straw WS and brown coal HKN is shown in Figure 5a for the major components (CO, $CO_2$, $H_2$, $CH_4$) and in Figure 5b for the minor gases, respectively.

Due to the higher oxygen content of WS, the proportion of oxygen containing compounds (CO and $CO_2$) in the pyrolysis gas was significantly higher compared to HKN (63 towards 47 vol.%), as expected. HKN released more $H_2$ and less molecular hydrocarbons ($C_2H_6$, $C_2H_4$, $C_3H_8$, $C_3H_6$) than wheat straw. Since the hydrogen contents in HKN and WS were similar, the behavior was caused by higher hydrogen amount releasing from WS involved in tar compounds.

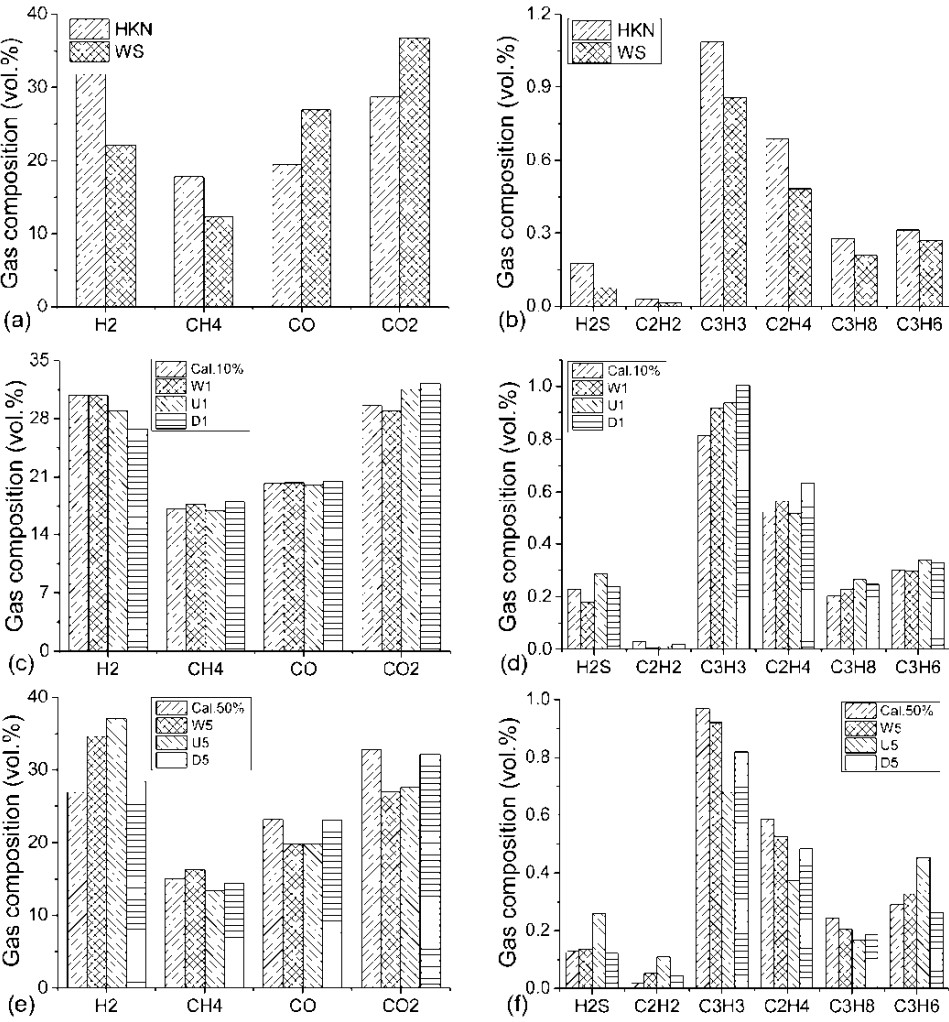

**Figure 5.** Experimental and calculated gas composition during co-pyrolysis of WS and HKN in LPA reactor, (**a**) major components from single fuel, (**b**) minor components from single fuel, (**c**) major components from 10 wt.% blends, (**d**) minor components from 10 wt.% blends, (**e**) major components from 50 wt.% blends, (**f**) minor components from 50 wt.% blends.



2) Gas composition from co-pyrolysis of HKN and WS

The gas composition resulting from the pyrolysis of the samples with 10 wt.% of WS addition at different contact conditions are shown in Figure 5c,d. For the major product gases, the yields of $H_2$ and $CO_2$ changed slightly at various positions, compared to the constant yields for CO and $CH_4$. The content of $H_2$ decreased and $CO_2$ increased, which corresponded to each other and was more significant for D10. The greatest differences in minor gas components ($C_2H_6$, $C_2H_4$ and $C_3H_6$) also occurred for D10. In LPA reactor, the flow direction of volatile with Argon was from top to bottom. The reactions possibly responsible for the synergy effects in case of samples with 10 wt.% of WS addition, can be illustrated as

$$D10: (V_{HKN} + P_{WS}) \times t_2 \ (t_1 > t_2)$$

$$U10: (V_{WS} + P_{HKN}) \times t_1 \ (t_1 > t_2)$$

where V is the produced volatiles and P is the symbol for the particles, while the indices WS and HKN stand for wheat straw and brown coal, respectively. As already observed in the TGA experiments, the longer contact time provided conditions supporting the occurrence of interactions. The more obvious synergy effect occurred for D10 sample showed that reactions between $V_{HKN}$ and $P_{WS}$ caused more interactions, even if the contact time of D10 was shorter than that of U10 in LPA. The reason responsible for synergy effects of D10 is that $V_{HKN}$ contained more $H_2$, most of which was produced at temperatures higher than 600 °C (the temperature for interaction occurred in TGA). $H_2$ can stabilize the free radical produced by primary pyrolysis to produce more gaseous products, so the contents of $C_2H_4$, $C_2H_6$, $C_3H_8$ and $C_3H_6$ were higher than expected. It is of note that the primary and secondary pyrolysis stages take place in parallel, and some of the primary volatiles released inside the particle can participate in secondary reactions to produce secondary products. This was the reason for reduction of $H_2$ content for 10 wt.% mixture samples. The increase of $CO_2$ content should be caused by the decrease of $H_2$, thus explaining why they correspond to each other so exactly. In addition, as shown in Table 1, the catalytic substances in HKN are mainly potassium and calcium, which could release at temperatures higher than 500 °C [31–33] to catalyze the decomposition of WS particles $P_{WS}$.

The gas products of 50 wt.% samples at different positions are illustrated in Figure 5e,f. Synergy effects occurred for W50 and U50, with increased $H_2$ yield and decreased $CO_2$ and CO yields, and these differed to interactions of 10 wt.% samples. The most obvious difference in gas composition occurred for the U50 sample. The reactions, which can cause synergy effects for 50 wt.% samples, showed as

$$D50: (V_{HKN} + P_{WS}) \times t_4 \ (t_3 < t_4)$$

$$U50: (V_{WS} + P_{HKN}) \times t_3 \ (t_3 < t_4)$$

The more obvious synergy effects occurred for U50 sample showed that reactions between $V_{WS}$ and $P_{HKN}$ cause more interactions (less contact time) compared to $V_{HKN}$ and $P_{WS}$. For higher $H_2$ yield, there were three possible explanations: (1) The water shift reaction produce $H_2$ by reducing CO. Water was a main component in biomass pyrolysis, which could react with carbon to produce $H_2$. (2) Sonobe et al. [14] reported that thermal decomposition of corncob was an exothermic process occurring between 250 and 475 °C. The produced heat can promote the cracking of straight and branched hydrocarbon chains and aromatic rings to produce $H_2$. (3) The catalytic mineral matter can promote reactions producing $H_2$ at high temperatures. Yin et al. [26] found that additions of Na and Ca both promoted $H_2$ production. Yang et al. [30] found the addition of K could enhance the production of $H_2$ at high temperatures, by promoting the water gas shift reaction. Therefore, as shown in Figure 4d, the Na, K, $H_2O$ and heat contained in volatiles of wheat straw ($V_{ws}$) can promote $H_2$ production by decomposition of brown coal particles ($P_{HKN}$) and the water shift reaction. The decrease of $CO_2$ should be caused by the increase of $H_2$.

To conclude, compared to contact time, reactions occurred between volatiles and particles from both fuels took more charge on synergy effects in LPA reactor. The amount of fuels used in LPA (20 gram)

was much more than in TGA (less than 1 gram), and there would be much more volatiles and catalytic compounds to promote the decomposition of WS and HKN. Therefore, unlike TGA, obvious synergy effects occurred for the blended samples with less contact time, since more produced volatiles can react with particles of the other fuel species along the gas flow direction, such as D10 and U50.

### 3.4. Volatile Release Kinetics (TGA) for Blend Samples

Table 3 illustrates the kinetic parameters, obtained by applying the Coast–Redfern method, for wheat straw, brown coal and blend samples with different particle contact conditions. The pyrolysis of blend samples showed two temperature ranges for pyrolysis, representing the wheat straw and the brown coal decomposition separately. The following results were obtained:

**Table 3.** Kinetics parameters for pyrolysis of single and blend samples under different contact conditions.

| Sample | Temperature (°C) | E (KJ/mol) | A (1/s) | $R^2$ |
|---|---|---|---|---|
| WS | 238–364 | 70.97 | $1.29 \times 10^5$ | 0.99 |
| HKN | 312–486 | 26.88 | $1.55 \times 10^2$ | 0.99 |
| W1 | 276–353 | 24.70 | $1.37 \times 10^2$ | 0.99 |
|  | 353–485 | 20.07 | $4.80 \times 10^{-1}$ | 1.00 |
| U1 | 269–350 | 26.41 | $1.86 \times 10^2$ | 0.99 |
|  | 350–480 | 20.04 | $4.30 \times 10^{-1}$ | 1.00 |
| D1 | 277–356 | 27.10 | $2.16 \times 10^2$ | 0.99 |
|  | 356–472 | 21.54 | $6.20 \times 10^{-1}$ | 1.00 |
| W5 | 267–360 | 51.40 | $1.07 \times 10^3$ | 1.00 |
|  | 360–474 | 5.24 | $1.61 \times 10^{-2}$ | 0.99 |
| U5 | 269–356 | 51.82 | $1.24 \times 10^3$ | 1.00 |
|  | 366–467 | 5.19 | $1.66 \times 10^{-2}$ | 1.00 |
| D5 | 271–355 | 52.55 | $1.41 \times 10^3$ | 1.00 |
|  | 365–463 | 5.74 | $2.03 \times 10^{-2}$ | 1.00 |

(1) For single pyrolysis, apparent activation energy E for biomass (70.97 KJ/mol) was almost three times higher than that of brown coal (26.88 KJ/mol). This was caused by much more catalytic compounds contained in HKN, including 6.34 wt.% of $Na_2O$, 37.51 wt.% of CaO and 10.13 wt.% of $Fe_2O_3$ in ash.

(2) For co-pyrolysis of biomass and brown coal, in wheat straw decomposition range, E values for blended samples were much lower than that for single WS pyrolysis, showing HKN addition could promote WS decomposition. For example, activation energy E in temperature range from 230 to 370 °C of U1 was 26.41 compared to 70.97 KJ/mol for WS pyrolyzed alone. In addition, the frequency factors A for all blend samples were much lower than that for single WS, indicating less collision between reactants. In particular, for 10 wt.% blend samples the order of magnitude for frequency factors A was $10^5$ for WS compared to $10^{-1}$ for blends. Therefore, the decreased reaction rate in the WS decomposition range was mainly caused by the heat and mass transfer, not by catalytic effects.

(3) In the brown coal decomposition range, E values for blended samples were lower compared to single HKN pyrolysis, showing that WS addition could promote the decomposition of HKN, and the catalytic capacity was much stronger for 50 wt.% samples. For example, activation energy E was 26.88 KJ/mol for HKN alone, around 20 KJ/mol for 10 wt.% samples and 5 KJ/mol for 50 wt.% samples. The frequency factor A for all blend samples in HKN decomposition range were also below single HKN pyrolysis, but that the effect was smaller than that in WS decomposition range. Therefore, the reason for the increased reaction rate at HKN decomposition range was mainly caused by the catalytic effect.

## 4. Conclusions

To investigate the effects of contact conditions between particles and volatiles on synergy effects during the co-pyrolysis process, the blends of wheat straw and brown coal were pyrolyzed under

different contact conditions (biomass was placed above and below coal, or they were well-mixed) in TGA and a fixed-bed reactor (LPA). The main results are summarized as follows:

(1) For TGA, the most intensive interactions in the reaction rate took place between 200 to 375 °C for all samples, with an obviously lower reaction rate than calculated value, due to the heat and mass transfer which can be certificated by kinetic parameters. At high temperatures (>600 °C), D10 (10 wt.% biomass was placed below coal) and U50 (50 wt.% biomass was placed above coal) showed obviously higher reaction rates, due to the longer contact time between particles and volatiles.

(2) For the LPA reactor, the char yield and catalytic composition (mainly potassium and calcium) in char for all blended samples was lower than calculated values. The amount of fuels used in LPA was much more than TGA, as there are many more volatiles and catalytic compounds that can promote the decomposition of WS and HKN. Therefore, unlike TGA, obvious synergy effects occurred for the blended samples with less contact time, since more produced volatiles could react with particles of the other fuel species along the gas flow direction, as for D10 and U50.

(3) It can be concluded that synergy effects in different reactors (TGA and fixed bed reactor) occurred by different mechanisms, due to different paths of volatiles flow, sample height, and so on. This provides important theoretical guidance for large commercial devices, since the reactor dimension in practice is considerably bigger than apparatus in experimental studies.

**Author Contributions:** Conceptualization, L.Z.; Data curation, G.Z. and L.Z.; Investigation, L.Z. and G.Z.; Project administration, B.M.; Supervision, D.K.

**Funding:** The authors are grateful to the financial support of the National Key Research and Development Program of China (2016YFB0600304) and National Natural Science Foundation of China (No. 51804313), the German Federal Ministry of Research and Education under research project "Deutsches Energie Rohstoffzentrum" (German Center for Energy raw materials, Project number: 03IS2021A), as well as the industry partners RWE Power AG, Mibra GmbH, Romonta and Vattenfall Europe Mining & Generation AG.

**Acknowledgments:** The authors are grateful to TU Bergakademie Freiberg, for providing the experimental facility.

**Conflicts of Interest:** The authors declare no conflict of interest.

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
