# Peer review of "Effects of Contact Conditions between Particles and Volatiles during Co-Pyrolysis of Brown Coal and Wheat Straw in a Thermogravimetric Analyzer and Fixed-Bed Reactor"

_processes, doi:10.3390/pr7040179_

Round 1
Reviewer 1 Report
Please find my detailed comments in the attached file.

Author Response
Thanks for your kindly advices. I have improved the paper according to your suggestion. The detailed response for each point was shown in the attachment.

Reviewer 2 Report
Aim of the paper is to investigate the synergy effect often observed during the co-pyrolysis of coal and biomass. In the study, a Rhenish lignite was chosen as representative of brown coal and wheat straw as biomass. The study was conducted in both thermogravimetric analyzer and in a laboratory scale fixed bed pyrolyzer. A kinetic analysis was also reported.
In the complex the paper is interesting, but, sometimes the given explanations seem a bit forced and not fully supported by the data.
About the TGA experiments observing figure 3 c-d, are evident the differences on the two peaks of the DTA. However, the other differences are very small almost negligible. The AA evidenced the higher weigh loss of the D10 and W10, but any comment is reported about this last sample in the follow of the paper. Again, the sample U10 show a lower conversion respect to the calculated value, but also in this case no explanation are reported. At page 8 line 267 the AA write: “The reaction rate of W10 and U10 was higher than the calculated value” but from the figure it is absolutely not evident, it seems that D10 has a higher reaction rate, while U10 and W10 has the same reaction rate than the calculated value.
About the given explanation (paragraph 3.2.3), it is convincing regarding the D10 and U50 samples, but it does not explain why U10 has lower conversions than the calculated value and why the sample W10 and W50 have a completely different behavior respect to the D10 and U50 respectively.
About Fixed bed experiments, table 2 reports the char analyses for all the samples, but probably for the samples U and D it would have been more useful to carry out the ultimate analyses on the two section of the sample (biomass and coal) separately. This would have better supported what the authors comments.
About the gas composition, because the intrinsically not steady process, and because the use of gas bags, there is some concern about the significance of the observed differences especially when these differences are very tiny. The AA need to explain the sampling strategy adopted: all the produced gas was collected in the bag and after an aliquot was analyzed or the samples were collected after a certain time?
In the following some minor comments
Abstract: please remove the acronyms D10 and U50 from the abstract. Use only a fully description of the acronym so that a reader can understand in what conditions the synergistic effect occurs. The same is true for the acronym LPA.
Introduction: pag 2 line 77, please insert here a reference to the Coats Redfern method.
Materials and Methods: The scheme of Figure 2 does not fit the description. It is not clear if the gas was sampled with the sample tube or with gas bags, and what is the need of the NaCl tank? Probably the scheme is referred to the whole apparatus that has also other capability. However , the scheme need to be modified to fit the actual use. Furthermore, it is necessary to specify the operative pressure used for the experiments (from the scheme it can’t work at atmospheric pressure!)
Author Response

(The authors gave the same response as above.)

Round 2
Reviewer 1 Report
The authors have addressed most of the comments; they have also tried to make changes according to the reviewer's suggestions. After revisions, the quality of the manuscript has been adequately enhanced. Therefore, the manuscript could be considered for the publication in the Journal. However, there are still some editing/ syntax errors presents in the manuscript which need to be corrected, hence publishing team is advised to read the manuscript carefully before publishing.
Author Response
Reviewer 1
The authors have addressed most of the comments; they have also tried to make changes according to the reviewer's suggestions. After revisions, the quality of the manuscript has been adequately enhanced. Therefore, the manuscript could be considered for the publication in the Journal. However, there are still some editing/ syntax errors presents in the manuscript which need to be corrected, hence publishing team is advised to read the manuscript carefully before publishing.
Answer: The grammar in the present paper has been carefully checked and the errors in the text have been corrected.

Reviewer 2 Report
The AA addressed almost all the referee comments and changed the manuscript in accordance with the suggestions. The paper is now more clear and complete. Only some minor aspects still need to be improved:
In line 24 of the abstract it is necessary to substitute the acronyms “LPA” with “fixed bed”
Lines 136-141 should be moved at the end of the paragraph (now line 144) because, otherwise, it is not completely clear what is the sample that will be heated to 800°C: if it is the collected sample or the sample in pyrolysis.
It is necessary to specify (at least in the caption of Figure 5) how is calculated the gas composition. In the description of the LPA apparatus it is reported: “…Then, all the gas bags were analyzed by a micro gas chromatograph (micro-GC) after experiment.” but, after how these analysis were used? The data reported in figure 5 are mean values or data collected at a certain time? This clarification is necessary for a clear understanding of results discussion.
Author Response
Reviewer 2
The AA addressed almost all the referee comments and changed the manuscript in accordance with the suggestions. The paper is now more clear and complete. Only some minor aspects still need to be improved:
In line 24 of the abstract it is necessary to substitute the acronyms “LPA” with “fixed bed”
Answer: The “LPA” was replaced by fixed bed reactor.
Lines 136-141 should be moved at the end of the paragraph (now line 144) because, otherwise, it is not completely clear what is the sample that will be heated to 800°C: if it is the collected sample or the sample in pyrolysis.
Answer: The sentence of “The samples were heated up from 25 to 800 °C with a heating rate of 10 K/min under argon atmosphere (50 ml/min STP).” was moved to the end of this paragraph.
It is necessary to specify (at least in the caption of Figure 5) how is calculated the gas composition. In the description of the LPA apparatus it is reported: “…Then, all the gas bags were analyzed by a micro gas chromatograph (micro-GC) after experiment.” but, after how these analysis were used? The data reported in figure 5 are mean values or data collected at a certain time? This clarification is necessary for a clear understanding of results discussion.
Answer: The system for gas collection consists of six glass bottles with scale and the volume of each bottle was 1000ml. The volume of gas products in each gas bag can be read by the scale on the bottle. By analysis of gas composition in each gas bag via a micro gas chromatograph, the total volume of certain gas (such as H2, CO2, CO) collected during the whole pyrolysis process can be calculated. Finally, the volume percent of certain gas can be obtained as shown in Fig. 5.
